# Loss of Frataxin induces iron toxicity, sphingolipid synthesis, and Pdk1/Mef2 activation, leading to neurodegeneration

Kuchuan Chen[1], Guang Lin[2], Nele A Haelterman[1], Tammy Szu-Yu Ho[3], Tongchao Li[1], Zhihong Li[2], Lita Duraine[4], Brett H Graham[1,2], Manish Jaiswal[2,4], Shinya Yamamoto[1,2,5], Matthew N Rasband[1,3], Hugo J Bellen[1,2,3,4,5]*

[1]Program in Developmental Biology, Baylor College of Medicine, Houston, United States; [2]Department of Molecular and Human Genetics, Baylor College of Medicine, Houston, United States; [3]Department of Neuroscience, Baylor College of Medicine, Houston, United States; [4]Howard Hughes Medical Institute, Baylor College of Medicine, Houston, United States; [5]Jan and Dan Duncan Neurological Research Institute, Texas Children's Hospital, Houston, United States

**Abstract** Mutations in *Frataxin (FXN)* cause Friedreich's ataxia (FRDA), a recessive neurodegenerative disorder. Previous studies have proposed that loss of *FXN* causes mitochondrial dysfunction, which triggers elevated reactive oxygen species (ROS) and leads to the demise of neurons. Here we describe a ROS independent mechanism that contributes to neurodegeneration in fly *FXN* mutants. We show that loss of *frataxin homolog (fh)* in *Drosophila* leads to iron toxicity, which in turn induces sphingolipid synthesis and ectopically activates 3-phosphoinositide dependent protein kinase-1 (Pdk1) and myocyte enhancer factor-2 (Mef2). Dampening iron toxicity, inhibiting sphingolipid synthesis by Myriocin, or reducing Pdk1 or Mef2 levels, all effectively suppress neurodegeneration in *fh* mutants. Moreover, increasing dihydrosphingosine activates Mef2 activity through PDK1 in mammalian neuronal cell line suggesting that the mechanisms are evolutionarily conserved. Our results indicate that an iron/sphingolipid/Pdk1/Mef2 pathway may play a role in FRDA.

*For correspondence: hbellen@bcm.edu

## Introduction

FRDA, an inherited recessive ataxia, is caused by mutations in *FXN* (*Campuzano et al., 1996*). During childhood or early adulthood, FRDA patients show a progressive neurodegeneration of dorsal root ganglia, sensory peripheral nerves, corticospinal tracts, and dentate nuclei of the cerebellum (*Koeppen, 2011*). *FXN* is evolutionarily conserved and the homologs have been identified in most phyla (*Bencze et al., 2006*; *Campuzano et al., 1996*). It encodes a mitochondrial protein that is required for iron-sulfur cluster assembly (*Layer et al., 2006*; *Lill, 2009*; *Muhlenhoff et al., 2002*; *Rotig et al., 1997*; *Yoon and Cowan, 2003*). Once synthesized, iron-sulfur clusters are incorporated into a variety of metalloproteins, including proteins of the mitochondrial electron transport chain (ETC) complexes and aconitase, where they function as electron carriers, enzyme catalysts, or regulators of gene expression (*Lill, 2009*). It has been proposed that loss of *FXN* leads to impaired ETC complex, which in turn triggers ROS production that directly contributes to cellular toxicity (*Al-Mahdawi et al., 2006*; *Anderson et al., 2008*; *Calabrese et al., 2005*; *Schulz et al., 2000*). However, the ROS hypothesis has been questioned in several studies. For example, loss of *FXN* only leads to a modest hypersensitivity to oxidative stress (*Macevilly and Muller, 1997*; *Seznec et al., 2005*; *Shidara and Hollenbeck, 2010*). In addition, several clinical trails based on antioxidant

**eLife digest** Friedreich's ataxia is a disorder in which nerve cells in the spinal cord, cerebellum and dorsal root ganglia progressively die as a person ages. People with this disorder often have difficulties with walking and can eventually develop other problems such as heart disease and diabetes. Mutations in a gene called *Frataxin,* known as *FXN* for short, are the primary cause of the disorder.

The *FXN* gene encodes a protein normally found in mitochondria – the structures that are best known for providing energy inside cells. Previous studies suggest that mutations in the *FXN* gene prevent mitochondria from working normally, which triggers the production of toxic chemicals called reactive oxygen species. However, therapies based on antioxidants (which combat reactive oxygen species) only have limited benefits in patients with Friedreich's ataxia; this suggests that other mechanisms contribute to the progression of the disease. Mutations in the *FXN* gene also cause iron to accumulate inside cells, which can be toxic too. However, it remains hotly debated whether or not iron toxicity contributes to Friedreich's ataxia.

Chen et al. set out to identify other mechanisms that can explain the loss of nerve cells seen in Friedreich's ataxia using fruit flies as an experimental system. Flies without the equivalent of *FXN* gene accumulated iron in their nervous systems and other tissues, but did not produce more reactive oxygen species. The experiments also revealed that this build-up of iron increased the production of fatty molecules (called sphingolipids), which in turn triggered the activation of two proteins (called Pdk1 and Mef2). Chen et al. then showed that blocking any of these effects could effectively delay the death of nerve cells in the mutant flies. Further experiments showed that boosting the levels of the Mef2 protein in the nerve cells of otherwise normal flies was enough to cause these cells to die.

The next step is to see whether the pathway also operates in mice and humans. Future studies could also see if dampening down this pathway could provide new treatments for Friedreich's ataxia.

therapy in FRDA patients have shown no or limited benefits (*Lynch et al., 2010*; *Parkinson et al., 2013*; *Santhera Pharmaceuticals, 2010*).

Loss of *yeast frataxin homolog* results in iron accumulation (*Babcock et al., 1997*), and this phenotype has also been reported in cardiac muscles of a *Fxn* deficiency mouse and FRDA patients (*Koeppen, 2011*; *Lamarche et al., 1980*; *Michael et al., 2006*; *Puccio et al., 2001*). However, whether iron accumulates in the nervous system upon loss of *FXN* remains controversial. Furthermore, whether iron deposits contribute to the pathogenesis is not clear. It has been reported that elevated iron levels were observed in the dentate nuclei or in glia cells of FRDA patients (*Boddaert et al., 2007*; *Koeppen et al., 2012*). Contrary to these results, others suggested that there is no increase of iron in the nervous system of *Fxn* deficiency mice and FRDA patients (*Koeppen et al., 2007*; *Puccio et al., 2001*; *Solbach et al., 2014*). Taken together, current data provide insufficient evidence to establish that iron dysregulation contributes to neurodegeneration. In addition, the mechanism underlying iron toxicity is still unclear. In summary, the pathological interplay of mitochondrial dysfunction, ROS, and iron accumulation remains to be established.

We identified the first mutant allele of *fh* in an unbiased forward genetic screen aimed at isolating mutations that cause neurodegenerative phenotypes. We show that loss of *fh* causes an age dependent neurodegeneration in photoreceptors and affects mitochondrial function. Unlike other mitochondrial mutants with impaired ETC activity, we do not observe an increase in ROS. However, loss of *fh* causes an iron accumulation in the nervous system, induces an up-regulation of sphingolipid synthesis, and activation of Pdk1 and Mef2. Reducing iron toxicity or inhibiting the sphingolipid/Pdk1/Mef2 pathway significantly suppresses neurodegeneration in *fh* mutants. To our knowledge, this is the first evidence that sphingolipids, Pdk1, and the transcription factor Mef2 are shown to play a primary role in FXN-induced neurodegeneration.

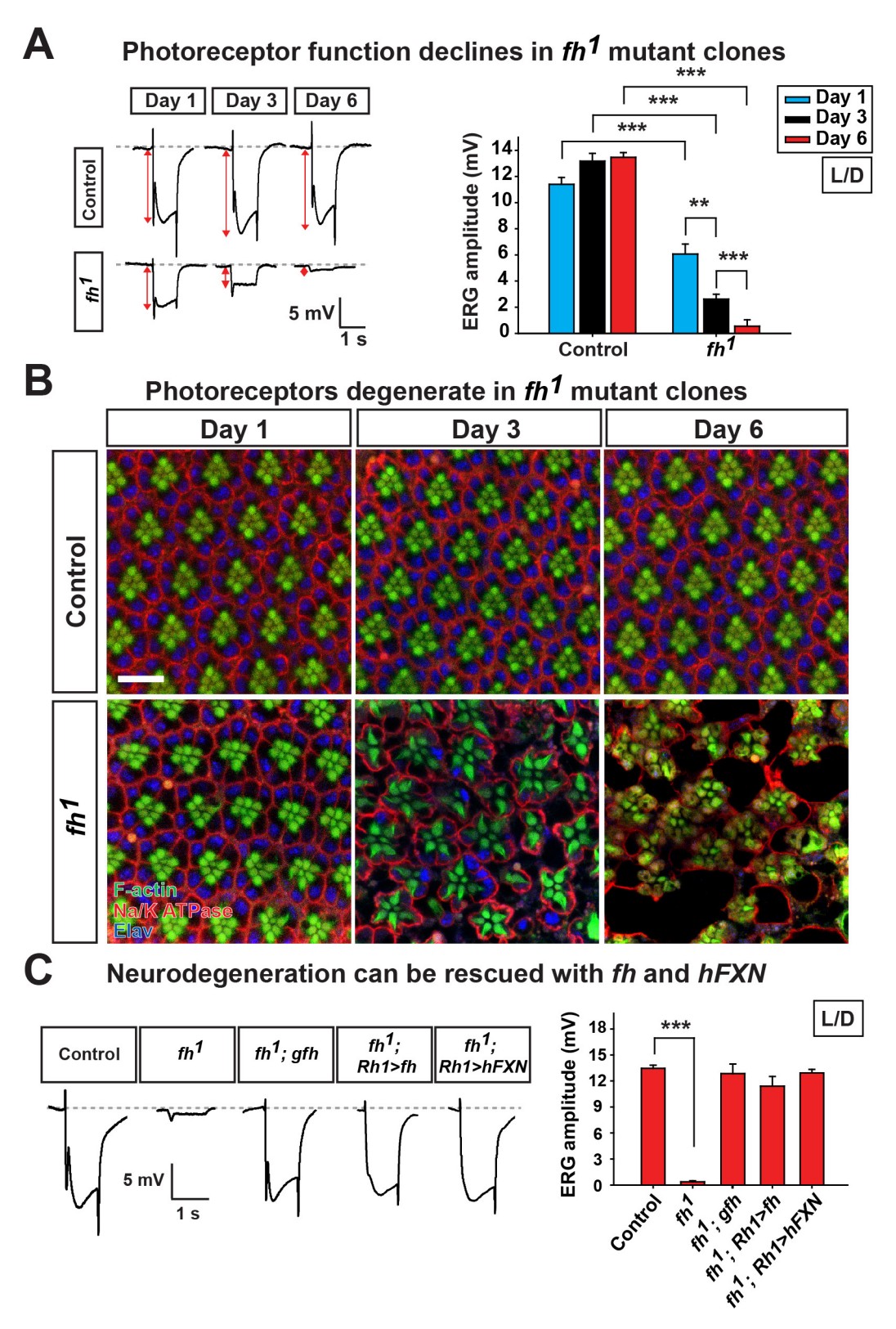

**Figure 1.** Loss of *fh* results in age dependent neurodegeneration in photoreceptors. (**A**) ERG of control (*y w FRT19A*) and *fh* (*y w fh FRT19A*) mosaic eyes. The ERG amplitudes from Day 1 to Day 6 are indicated by red double arrows. Quantification of the ERG amplitudes of control (*y w FRT19A*) and *fh* (*y w fh FRT19A*) is on the right. (**B**) Retina morphology of control (*y w FRT19A*) and *fh* mosaic eyes. Rhabdomeres are labeled by phalloidin (green),

*Figure 1 continued*

whereas anti-Na/K ATPase antibody (red) marks the photoreceptor membranes. Neuronal nuclei are labeled by anti-Elav antibody (blue). Scale bar: 10 μm. (C) ERG of control (*y w FRT19A*), *fh* mutants, *fh* mutants carrying a genomic *fh* construct (*fh; gfh*), and *fh* mutants that express the *fh* cDNA (*fh; Rh1>fh*) or human *FXN* cDNA (*fh; Rh1>hFXN*) in photoreceptors using a *Rh1-GAL4* driver. The ERG amplitudes were recorded at Day 6. Data are presented as mean ± SEM. **p<0.01, ***p<0.001, Student's t-test. L/D, flies are raised in 12 hr light and dark cycle.

The following figure supplements are available for figure 1:

**Figure supplement 1.** Mutation and lethal phase analysis of *fh* mutants.

**Figure supplement 2.** Ultrastructure of adult eyes exhibits morphological defects in *fh* mutant clones.

## Results

### *fh* mutants display an age dependent neurodegeneration of photoreceptors

We identified the first EMS (ethyl methanesulfonate) induced missense mutation in *fh*, the *Drosophila* homolog of *FXN*, through a forward genetic screen on the X chromosome (*Figure 1—figure supplement 1A*) (*Haelterman et al., 2014*; *Yamamoto et al., 2014*). The molecular lesion, S136R, is in a highly conserved region, which is required for the *FXN* binding to the iron-sulfur cluster assembly complex (*Tsai et al., 2011*). In addition, six point mutations adjacent to S136 (S157 in human) have been reported in FRDA patients (*Figure 1—figure supplement 1B*) (*Santos et al., 2010*). Hemizygous *fh* mutants are L3 to pupal lethal, and the lethality is not enhanced in transheterozygous mutants that carry a deficiency (*fh/Df (1)BSC537*), suggesting that S136R is a severe loss-of-function (*Figure 1—figure supplement 1C*). The lethality of *fh* can be rescued by a genomic *fh* construct (*gfh*), or by ubiquitous expression (*da-GAL4* driver) of *fh* cDNA using the UAS/GAL4 system (*Brand and Perrimon, 1993*) (*Figure 1—figure supplement 1C*). The lethality cannot be rescued with tissue specific drivers, including neuronal (*nSyb-GAL4*), muscular (*Mef2-GAL4*), or glial (*repo-GAL4*) drivers, suggesting a requirement for *fh* in multiple tissues (*Figure 1—figure supplement 1C*). In addition, *fh* mutants exhibit a smaller body size and prolonged larval stages (10 to 12 days instead of 5 days for wild type animals), similar to other mitochondrial mutants (*Sandoval et al., 2014*; *Zhang et al., 2013*). Interestingly, removal of maternal wild type *fh* mRNA or protein in the egg exacerbates the lethal phase from pupal to embryonic lethality (*Figure 1—figure supplement 1C*). In essence, the residual maternally deposited wild type Fh not only extends the lifespan but also creates a partial loss of Fh condition in the *fh* mutants.

To bypass pupal lethality of *fh* mutants, mosaic mutant clones of adult photoreceptor neurons were generated with the *eyeless*-FLP/FRT system. As a functional readout, we recorded and compared electroretinograms (ERGs) in young and aged flies. In response to a light stimulus, the ERG amplitudes of newly eclosed *fh* mutants (Day 1) are ~60% of control animals (*y,w,FRT19A*$^{iso}$, the isogenized flies used in the screen) (*Figure 1A*). The ERG amplitudes rapidly decline over a six day period (*Figure 1A*). Morphologically, the *fh* mutant clones contain a normal number of photoreceptors with a typical trapezoidal organization at Day 1 (*Figure 1B*). However, by using transmission electron microscopy (TEM), we found that Day 1 *fh* mutant photoreceptors exhibit an expansion of the endoplasmic reticulum (ER) and a dramatic accumulation of lipid droplets in glial cells that are not observed in controls (*Figure 1—figure supplement 2A and B*). At Day 3, photoreceptors in mutant clones exhibit aberrant elongated rhabdomeres (*Figure 1B*). By Day 6, many of the photoreceptors are missing in the mutant clones, whereas control retinas exhibit intact morphology (*Figure 1B* and *Figure 1—figure supplement 2C*). These data show that the mutant photoreceptors undergo a rapid and severe functional as well as morphological degeneration.

The neurodegeneration can be fully rescued by a genomic *fh* construct or by expressing *fh* cDNA in photoreceptors (*Figure 1C* and *Figure 1—figure supplement 2D*), suggesting that photoreceptor degeneration is cell-autonomous. Moreover, expression of the human *FXN* cDNA rescues neurodegeneration in *fh* mutants (*Figure 1C* and *Figure 1—figure supplement 2D*), showing that fly and human *FXN* have conserved molecular function, and that our studies are relevant to the biological role of human *FXN*.

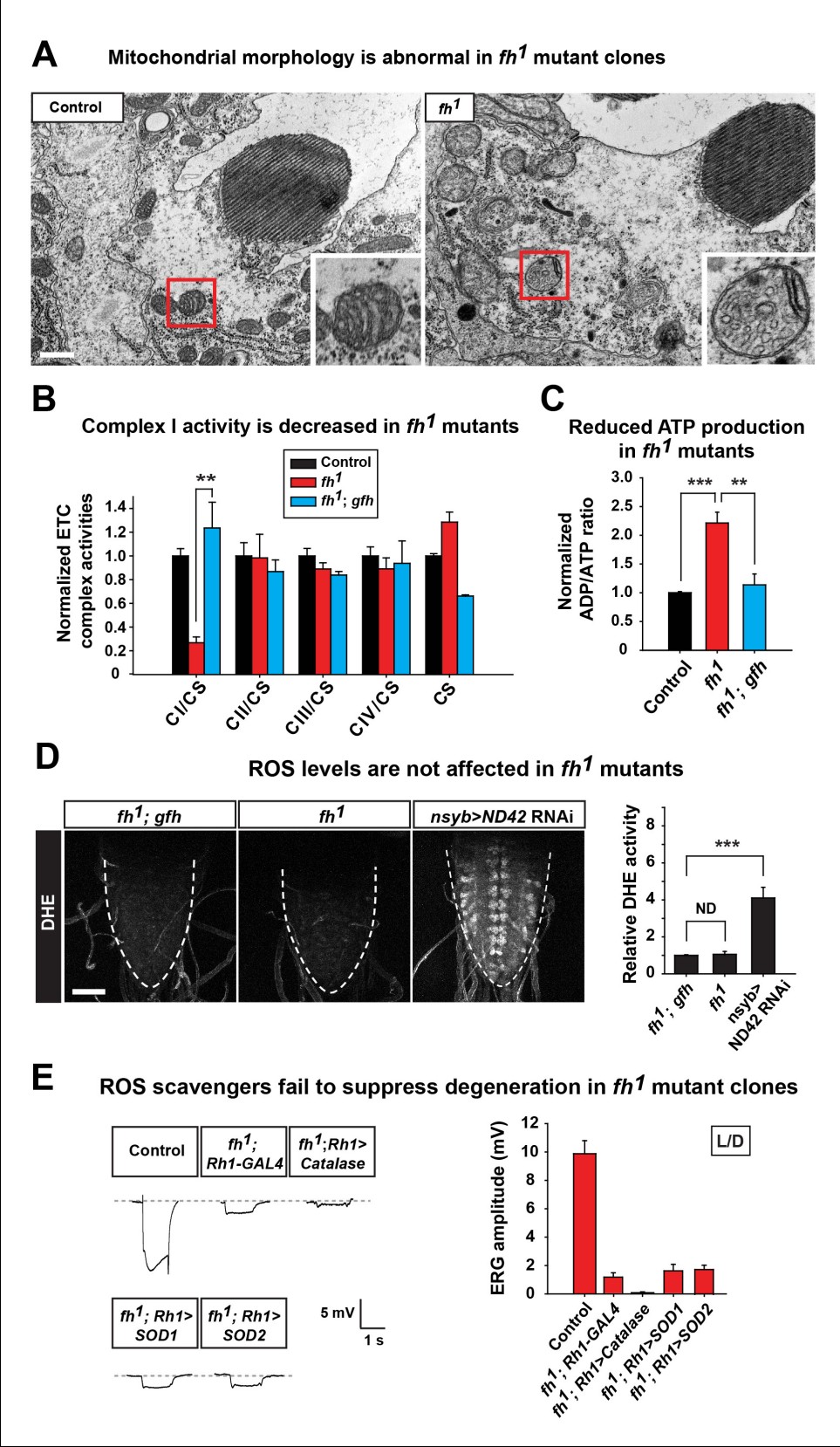

**Figure 2.** Loss of *fh* causes impaired mitochondria, and reducing ROS levels fails to suppress degeneration in *fh* mutants. (**A**) TEM images of mitochondria in photoreceptors of control (*y w FRT19A*) and *fh* mutant clones. The *Figure 2 continued on next page*

*Figure 2 continued*

insets show a single mitochondrion (red box). Scale bar: 500 nm. (B) ETC complex activities in control (*y w FRT19A*), *fh* mutant, and genomic rescued animal (*fh; gfh*). Complex activities are normalized to citrate synthase (CS), which act as a control of mitochondrial mass. n = 3. (C) ADP/ATP ratio in control (*y w FRT19A*), *fh* mutant, and rescued animal (*fh; gfh*). n = 3. (D) ROS levels are measured by DHE in the larval ventral nerve cord. Neuronal down-regulation of ND42 (*nSyb>ND42 RNAi*), a subunit of CI, acts as a positive control. Quantification of DHE fluorescence is on the right. n = 8. Scale bar: 50 μm. (E) Day 3 ERG of control (*y w FRT19A; Rh1-GAL4*) and *fh* mutants in which ROS scavengers are overexpressed in photoreceptors. Data are presented as mean ± SEM. **p<0.01, ***p<0.001, Student's t-test. ND, no significant difference. L/D, flies are raised in 12 hr light and dark cycle.

The following figure supplements are available for figure 2:

**Figure supplement 1.** Mitochondrial phenotype of *fh* mutants.

**Figure supplement 2.** ROS/JNK/SREBP pathway does not contribute to degeneration in fh mutants.

## Mitochondria are dysfunctional but ROS levels are not altered in *fh* mutants

FXN has been reported to be localized to mitochondria (*Koutnikova et al., 1997*). Indeed, Fh colocalizes with mitoGFP in larval motor neuron, and removing the predicted mitochondrial targeting sequence (MTS) of *fh* cDNA (Fh-ΔMTS-V5) leads to a diffused signal in the cytosol (*Figure 2—figure supplement 1A*). Removal of the MTS leads to a dysfunctional protein as expression of Fh-ΔMTS-V5 in the *fh* mutant clone is not able to rescue the ERG defect (*Figure 2—figure supplement 1B*).

Loss of *fh* causes a highly aberrant mitochondrial morphology in one day old adult photoreceptors. The mitochondria in *fh* mutants are larger and exhibit aberrant cristae and altered inner membrane morphology (*Figure 2A* and *Figure 2—figure supplement 1C*). Loss of *FXN* has been reported to cause various ETC and respiratory defects in different animal models, including yeast, fly, mouse, as well as FRDA patients (*Al-Mahdawi et al., 2006*; *Anderson et al., 2005*; *Carletti et al., 2014*; *Koutnikova et al., 1997*; *Puccio et al., 2001*; *Rotig et al., 1997*; *Wilson and Roof, 1997*). We find that ETC Complex I (CI) activity is severely reduced in *fh* mutants when compared to control and genomic rescued animals (*Figure 2B*). Consistent with this data, the ADP/ATP ratio is increased in *fh* mutants (*Figure 2C*), showing that energy production is impaired.

It has been reported that expression of yeast Ndi1p, which is a single subunit of NADH dehydrogenase, can mitigate the deleterious effects of an ETC CI deficit in the fly (*Vilain et al., 2012*). However, overexpression of Ndi1p in *fh* mutant clones does not suppress neurodegeneration (*Figure 2—figure supplement 1D*). Impaired ETC CI activity has been linked to increased ROS production that in turn causes cellular stress or toxicity (*Sugioka et al., 1988*; *Turrens and Boveris, 1980*). We therefore evaluated ROS production in vivo with dihydroethidium (*DHE*), a dye that emits fluorescence upon oxidation by ROS. Neuronal down-regulation of ND42 (*nSyb>ND42 RNAi*), a subunit of CI, was used as a positive control (*Owusu-Ansah et al., 2008*). As shown in *Figure 2D*, loss of ND42 exhibits strong DHE fluorescence in the larval ventral nerve cord, indicating elevated levels of ROS. However, DHE levels are not obviously increased in *fh* mutants, suggesting no or a very mild increase in ROS (*Figure 2D*).

Elevated ROS levels are proposed to be a major contributor to the pathogenesis in FRDA. To further investigate the role of ROS in photoreceptor degeneration of *fh* mutants, we overexpressed several ROS scavengers, including *Catalase, Super Oxide Dismutase 1 (SOD1)*, and *SOD2*, which have all been shown to effectively reduce ROS in flies (*Orr and Sohal, 1992*; *Parkes et al., 1998*; *Sun et al., 2002*). Consistent with the DHE data, overexpression of these ROS scavengers does not suppress neurodegeneration in *fh* mutants (*Figure 2E*). Furthermore, feeding *fh* mutants with AD4, a potent antioxidant that dampens oxidative stress (*Amer et al., 2008*; *Liu et al., 2015*), does not suppress neurodegeneration (*Figure 2—figure supplement 1E*). Taken together, our findings argue that ROS does not contribute to the demise of neurons in *fh* mutants.

It has been proposed that lipid droplets play a role in the pathogenesis of FRDA as knockdown of *fh* in flies causes lipid droplet accumulation in glia (*Navarro et al., 2010*). From TEM and nile red

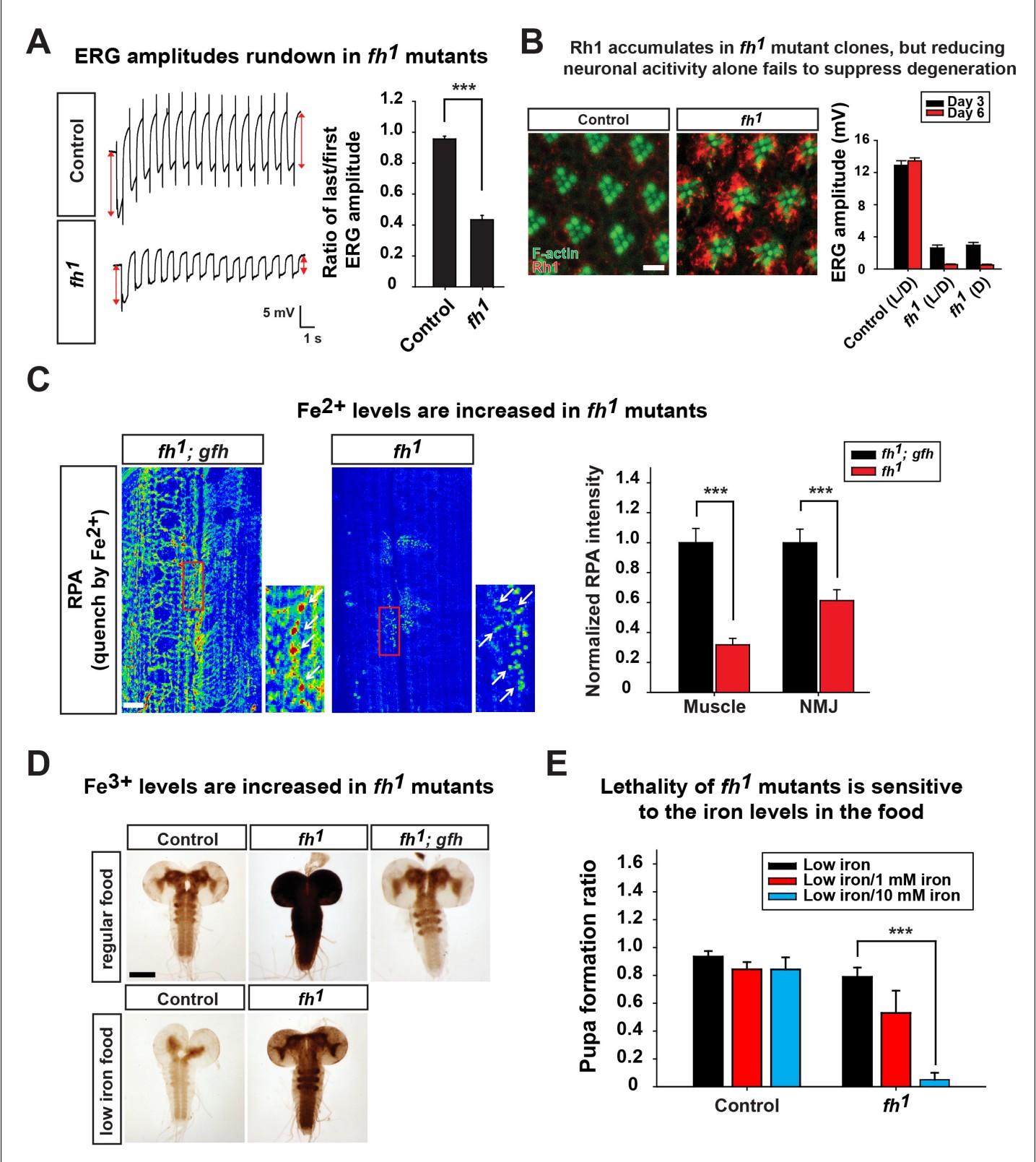

**Figure 3.** Loss of *fh* causes Rh1 accumulation and iron deposit. (A) ERG traces upon repetitive light stimulation in both control (*y w FRT19A*) and *fh* mutants. The double red arrows indicate ERG amplitudes of the first and last trace during the stimulation. Quantification of the ratio of last/first ERG amplitude is on the right. (B) Rh1 distribution in control (*y w FRT19A*) and *fh* mutant clones after a 12 hr light exposure. Rh1 is labeled in red, and

*Figure 3 continued on next page*

*Figure 3 continued*

rhabdomeres are labeled in green. Scale bar: 5 μm. Quantification of ERG amplitudes of control (*y w FRT19A*) and *fh* mutant clones in different light conditions is on the right. L/D, flies are raised in 12 hr light and dark cycle. D, flies are raised in dark. (C) $Fe^{2+}$ levels are assessed by RPA in larval muscle and NMJ in *fh* mutants and rescued animals (*fh; gfh*). RPA fluorescence is represented as a heat map. The NMJ boutons are indicated by white arrows. Quantification of normalized RPA intensity is on the right. n = 8. Scale bar: 20 μm. (D) Larval brain Perls'/DAB staining of control (*y w FRT19A*), *fh* mutants, and rescued animals (*fh; gfh*) with regular food or low iron food conditions. Scale bar: 100 μm. (E) Percent of animals that develop into pupae in control (*y w FRT19A*) and *fh* mutants when animals are raised by low iron food with different iron concentration. n = 3. Data are presented as mean ± SEM. ***p<0.001, Student's t-test.

The following figure supplement is available for figure 3:

**Figure supplement 1.** $Fe^{2+}$ staining control.

staining, we observed a strong lipid droplet accumulation in glial cells of *fh* mutant clones (*Figure 1—figure supplement 2A* and *Figure 2—figure supplement 2A*). We recently reported that similar phenotype is observed in several mitochondrial mutants and promotes neurodegeneration via a ROS/JNK/SREBP pathway (*Liu et al., 2015*) (*Figure 2—figure supplement 2B*). However, we do not observe elevated ROS levels (*Figure 2D*), and the level of a JNK pathway reporter, *puc-lacZ*, is not altered in *fh* mutants (*Figure 2—figure supplement 2C*). Finally, overexpression of *brummer lipase (Bmm)* or *lipase 4 (lip4)*, homologs of human Adipose Triglyceride Lipase (ATGL) or acid lipase (*Gronke et al., 2005*) respectively, effectively reduces the number of lipid droplet but does not delay neurodegeneration of *fh* mutants (*Figure 2—figure supplement 2A*), indicating that these lipid droplets do not contribute to neurodegeneration.

## Dysfunctional mitochondria cause Rhodopsin1 accumulation in *fh* mutants

We recently discovered that some mutants that are required for mitochondrial function display an activity dependent degeneration of photoreceptors (*Jaiswal et al., 2015*). These mitochondrial mutants share several common phenotypes, including reduced ATP production, ERG rundown upon repetitive stimulation, and Rhodopsin1 (Rh1) accumulation in the photoreceptor cytoplasm upon light exposure (*Jaiswal et al., 2015*). In these mutants, dysfunctional mitochondria cause reduced ATP production and decreased calcium influx, which not only leads to reduced photoreceptor depolarization, but also triggers an internalization of Rh1 leading to an overload of the endolysosomal system upon light stimulation (*Jaiswal et al., 2015*). The accumulation of Rh1 results in activity dependent degeneration, as the photoreceptors only degenerate when flies are raised in light (light-induced neuronal activity) but not in dark (no neuronal activity) (*Alloway et al., 2000*; *Chinchore et al., 2009*; *Jaiswal et al., 2015*).

Since loss of *fh* leads to decreased CI activity and reduced energy production (*Figure 2B and C*), we hypothesized that a similar activity dependent degeneration mechanism contributes to neurodegeneration in *fh* mutants. To test this hypothesis, we first examined whether ERG traces rundown upon repetitive light stimulation in *fh* mutant photoreceptors. As shown in *Figure 3A*, the ERG amplitudes of the first and last traces show a similar size in control animals. In contrast, the ERG amplitude rapidly declines during repetitive light stimulation in *fh* mutants (*Figure 3A*), similar to other mitochondrial mutants (*Jaiswal et al., 2015*). Therefore, we assessed whether Rh1 internalization is affected in *fh* mutants. Wild type flies exposed to 12 hr of constant light typically exhibit low levels of Rh1 in photoreceptors (*Figure 3B*). However, in *fh* mutants, Rh1 accumulates in the cytoplasm of photoreceptors (*Figure 3B*). Since *fh* mutant photoreceptors share several features with other mitochondrial mutants, we hypothesized that reducing photoreceptor activity by raising *fh* mutants in the dark would suppress degeneration. However, *fh* mutant flies that are raised in the dark still display severe photoreceptor degeneration (*Figure 3B*). Hence, we propose that in addition to the Rh1 accumulation, another pathogenic mechanism must contribute to neurodegeneration and mask the rescue effect when flies are raised in dark.

## Iron accumulates in the nervous system of *fh* mutants

Whether iron accumulates in the nervous system and contributes to the pathogenesis in mice and FRDA patients has remained a topic of debate (*Boddaert et al., 2007*; *Puccio et al., 2001*; *Solbach et al., 2014*). We therefore investigated whether iron is involved in neurodegeneration of *fh* mutants. To assess if iron homeostasis is affected in *fh* mutants, we first examined ferrous iron ($Fe^{2+}$) levels by Rhodamine B-((1,10-phenanthrolin-5-yl)-aminocarbonyl) benzyl ester (RPA). RPA is a cell-permeable fluorescent dye that selectively accumulates in mitochondria, and its fluorescence is stoichiometrically quenched by $Fe^{2+}$ (*Petrat et al., 2002*). As RPA poorly crosses the blood brain barrier (data not shown), we performed RPA staining of larval muscles and neuromuscular junctions (NMJ). In *fh* mutants rescued with the genomic *fh*, which serves as a negative control, the RPA signal exhibits strong fluorescence (*Figure 3C*). In contrast, the RPA intensity in *fh* mutants is dramatically reduced, in both muscle and NMJ boutons, suggesting a strong $Fe^{2+}$ accumulation in mitochondria (*Figure 3C*). RPAC (Rhodamine B-((phenanthren-9-yl)-aminocarbonyl) benzyl ester), a dye that is structurally very similar to RPA, is also taken up by mitochondria but is insensitive to $Fe^{2+}$ mediated fluorescence quenching. Indeed, RPAC fluorescence is not quenched in rescued animals (*fh; gfh*) and *fh* mutants (*Figure 3—figure supplement 1*), suggesting that the decreased fluorescence of RPA in *fh* mutants is not due to impaired dye uptake.

To investigate whether ferric iron ($Fe^{3+}$) levels are increased in *fh* mutants, we performed Perls' staining and used 3,3'-*Diaminobenzidine* (*DAB*) to enhance the signal (*Meguro et al., 2007*). In control and rescued animals, the DAB signal is most obvious in the neuropil of the ventral nerve cord (*Figure 3D*), suggesting that $Fe^{3+}$ is unevenly distributed in the nervous system. In *fh* mutants, however, the DAB signal is dramatically increased (*Figure 3D*), indicating that loss of *fh* leads to a severe accumulation of $Fe^{3+}$. Increased $Fe^{3+}$ levels are not only found in the larval brain, but can also be observed in fat body and gut (data not shown). Together, these results indicate that loss of *fh* leads to both $Fe^{2+}$ and $Fe^{3+}$ accumulation in multiple tissues, including the nervous system.

## Iron toxicity cause neurodegeneration in *fh* mutants

To determine if the iron accumulation contributes to neurodegeneration in *fh* mutant clones, we reduced iron levels in the fly food. Our regular food contains iron-rich molasses. To reduce iron levels, we replaced molasses with sucrose as the main carbohydrate source and refer to this food as 'low iron food'. As shown in *Figure 3D*, feeding larvae with low iron food reduces iron levels in the brains of both wild type control and *fh* mutants. In addition, the lethal phase of *fh* mutants is susceptible to iron levels in the food. In low iron food, 80% of *fh* mutants develop into pupae, but addition of 10 mM $Fe^{3+}$ to the low iron food reduces pupation to about 10%, whereas the pupation rate of controls is not affected (*Figure 3E*). These data indicate that *fh* mutants are highly sensitive to iron levels. To determine if iron toxicity is underlying the observed neurodegeneration, we fed low iron food to *fh* mutants and examined photoreceptor degeneration. Treating *fh* mutants with low iron food improves the ERG defects when compared to regular food condition (*Figure 4A*), suggesting that iron mediated toxicity contributes to neurodegeneration in *fh* mutants.

## Neuronal activity and iron synergize to promote neurodegeneration

We next assessed if excess neuronal activity and iron accumulations act synergistically to promote neurodegeneration in *fh* mutants. Interestingly, reducing photoreceptor activity (raising flies in dark) and iron (low iron food) leads to a synergistic effect in restoring both photoreceptor function and morphology (*Figure 4A and B*). To exclude the possibility that changes in other compounds than iron in low iron food suppresses neurodegeneration in *fh* mutants, we reintroduced iron into the low iron food (low iron/10 mM iron) and found that this treatment again induces rapid degeneration in *fh* mutant clones even when animals are raised in dark (*Figure 4B and C*). The ERG amplitudes of the wild type controls are not affected by different iron concentrations in the food (*Figure 4—figure supplement 1*). These data suggest that iron toxicity contributes to the activity independent degeneration, as photoreceptors degenerate in the absence of neuronal activity. We also tested if feeding 100 mM iron to wild type controls is sufficient to trigger neurodegeneration. This concentration of iron is toxic as most animals die as embryos or young larval instars, and the $Fe^{3+}$ levels were not increased in the brains (data not shown).

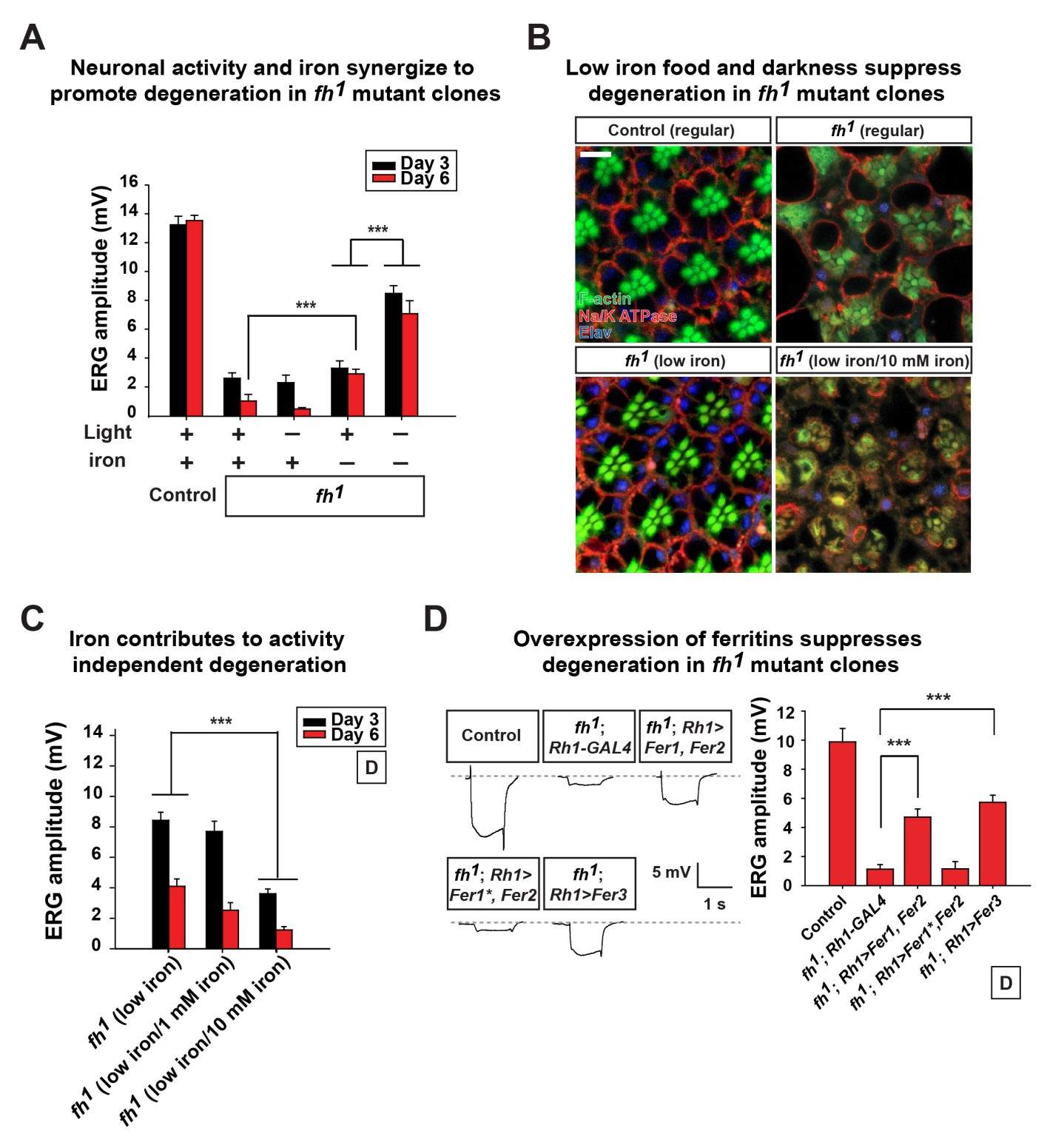

**Figure 4.** Neuronal activity and iron interact synergistically to contribute to photoreceptor degeneration in *fh* mutants. (A) Quantification of ERG amplitudes of control (*y w FRT19A*) and *fh* mutants under different light and iron conditions. (B) Retina morphology of control (*y w FRT19A*) and *fh* mutants under different food conditions at Day 6. All animals are raised in dark. Regular, flies are raised in regular food. Low iron, flies are raised in low iron food. Scale bar: 5 μm. (C) Quantification of ERG amplitudes of *fh* mutants in low iron food with different iron concentrations. (D) ERG of control (*y*

*Figure 4 continued on next page*

*Figure 4 continued*

*w FRT19A; Rh1-GAL4*) and *fh* mutants in which ferritins are overexpressed. *Fer1*\*, enzymatic dead form of *Fer1HCH*. Data are presented as mean ± SEM.
***p<0.001, Student's t-test.
The following figure supplement is available for figure 4:

**Figure supplement 1.** Control of low iron food treatment.

We next explored if genetic manipulations can suppress iron toxicity and neurodegeneration in *fh* mutants. We expressed Ferritins, the iron storage proteins that chelate and sequester irons. In *Drosophila*, *Ferritin 1 heavy chain homologue* (*Fer1HCH*) and *Ferritin 2 light chain homologue* (*Fer2LCH*) form a heteropolymeric complex, whereas *Ferritin 3 heavy chain homologue* (*Fer3HCH*) forms a homopolymeric complex to store irons (*Missirlis et al., 2006*, *2007*). In *fh* mutant clones, co-expression of *Fer1HCH* and *Fer2LCH* (*fh; Rh1>Fer1, Fer2*) or expression of *Fer3HCH* alone (*fh; Rh1>Fer3*) significantly suppresses degeneration when flies are raised in dark (*Figure 4D*). Co-expression of a ferroxidase inactive form of *Fer1HCH* along with wild type *Fer2LCH* in *fh* mutants (*fh; Rh1>Fer1*\*, *Fer2*) did not rescue degeneration (*Figure 4D*). These results further confirm that iron toxicity contributes to neurodegeneration in *fh* mutant clones. In summary, we conclude that excess neuronal activity (activity dependent) and accumulated iron (activity independent) synergize to promote photoreceptor degeneration in *fh* mutants.

## Increased sphingolipid synthesis mediates iron toxicity and causes neurodegeneration

Given that iron accumulates in *fh* mutants, we investigated the mechanism downstream of iron toxicity. Previous studies proposed that highly active iron interacts with hydrogen peroxide to generate free radicals, the Fenton reaction. However, we did not observe an increase in ROS (*Figure 2D*), and expression of different ROS scavengers did not suppress degeneration in *fh* mutants (*Figure 2E*). It has been previously documented that wild type yeast grow in medium with high iron levels increases sphingolipid synthesis (*Lee et al., 2012*). However, this phenotype was not documented in the *yeast frataxin homolog* mutants or *FXN* mutants in other organisms. As we observed a strong iron accumulation in *fh* mutants, we hypothesized that iron toxicity induces sphingolipid synthesis and causes neurodegeneration.

To test our hypothesis, we first assessed the *de novo* synthesis of sphingolipids by mass spectrometry. Consistent to our hypothesis, several upstream metabolic intermediates of sphingolipids are increased in *fh* mutants (*Figure 5A*). Strikingly, dihydrosphingosine (dhSph) is increased more than 10 fold in *fh* mutants. Other sphingolipids, such as dihydroceramide (dhCer) and sphingosine (Sph), are also increased to different levels (*Figure 5A*). This data suggest that the *de novo* synthesis of sphingolipids is increased in *fh* mutants.

To determine if increased sphingolipid synthesis contributes to neurodegeneration in *fh* mutants, we performed RNAi experiments against the first and rate-limiting enzyme in the *de novo* synthesis pathway, *lace,* the fly homolog of *Serine palmitoyltransferase*. This significantly improved the ERG defects of *fh* mutants (*Figure 5B*). In addition, feeding *fh* mutants Myriocin, a serine palmitoyltransferase inhibitor, suppressed both the functional and morphological photoreceptor defects of *fh* mutant clones (*Figure 5C*). Feeding myriocin to control animals does not adversely affect photoreceptor function (*Figure 5—figure supplement 1*). To assess if iron accumulation and increased sphingolipid synthesis act in the same or a different pathway, we treated *fh* mutants with low iron food as well as myriocin. As shown in *Figure 5D*, we did not observe an additive or synergistic effect. These results suggest that iron toxicity induces sphingolipid synthesis which in turn causes neurodegeneration in *fh* mutants.

## Sphingolipids trigger the Pdk1/Mef2 pathway and cause neurodegeneration in *fh* mutants

Increased sphingolipids in yeast have been shown to activate a kinase pathway (Pkh1/Ypk1/Smp1) that converges on a transcription factor Smp1p (*Lee et al., 2012*). However, this pathway has not yet been studied in higher eukaryotic cells, or *FXN* mutant animal models. The corresponding

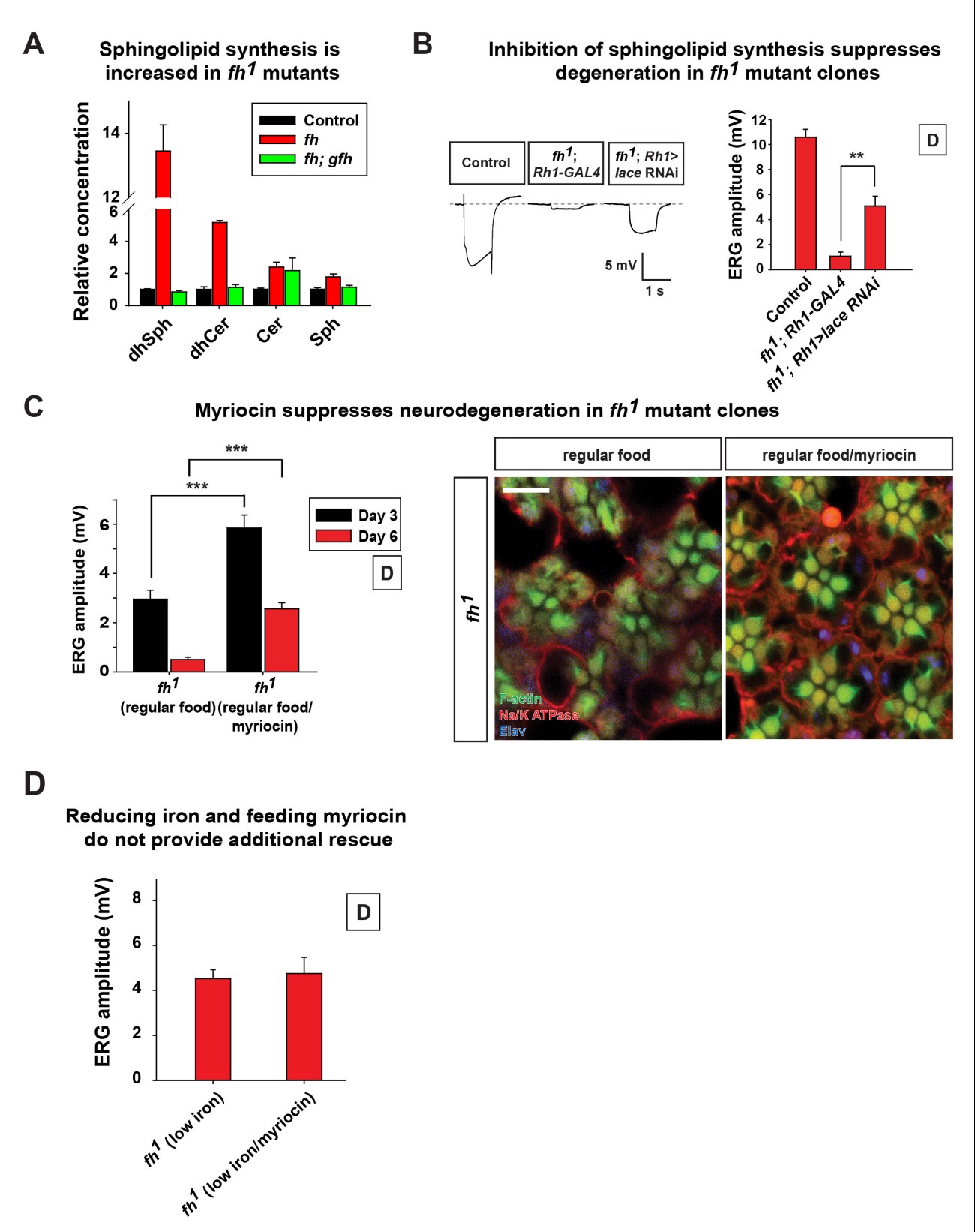

**Figure 5.** Increased sphingolipids contribute to degeneration of *fh* mutants. (A) Mass spectrometry analysis of sphingolipids in control (*y w FRT19A*), *fh* mutants, and rescued animals (*fh; gfh*). dhsph, dihydrosphingosine; dhCer, dihydroceramide; Cer, ceramide; Sph, sphingosine. n = 2. (B) ERG of control

*Figure 5 continued on next page*

*Figure 5 continued*

(*y w FRT19A; Rh1-GAL4*), and *fh* mutants in which an RNAi against *lace* is expressed. (C) ERG amplitude and retinal morphology of *fh* mutants treated with 100 μM myriocin. Scale bar: 5 μm. (D) Quantification of Day 3 ERG amplitudes of *fh* mutants with low iron food and myriocin treatments. Data are presented as mean ± SEM. The error bars represent the range of data in (A). **p<0.01, ***p<0.001, Student's t-test. D, flies are raised in dark.
The following figure supplement is available for figure 5:

**Figure supplement 1.** Control of myriocin treatment.

proteins in mammals are PDK1, SGK (*serum-* and glucocorticoid regulated kinase), and Mef2 transcription factor (*Casamayor et al., 1999*; *Dodou and Treisman, 1997*). The *Drosophila* homolog of *SGK* has not been identified, but both *Pdk1* and *Mef2* are present in the fly. *Mef2* is a well known transcription factor required for muscle differentiation, however, it is also expressed in certain neurons and photoreceptors (*Blanchard et al., 2010*). As we observed an increase in sphingolipids, we assessed if increased levels of sphingolipids activate the Pdk1/Mef2 pathway and underlie the neurodegenerative phenotype in *fh* mutants. We first tested if Pdk1/Mef2 is activated in *fh* mutants. As a proxy for Mef2 activation, we examined the mRNA levels of known downstream targets of Mef2. Seven Mef2 targets, which have been validated through in situ hybridization and qRT-PCR, are all ectopically expressed in response to ectopic expression of Mef2 in vivo (*Elgar et al., 2008*). We therefore examined these seven Mef2 targets in the nervous system of *fh* mutants. As shown in *Figure 6A*, all of these Mef2 targets are up-regulated. Several negative control genes, the downstream targets of other transcription factors, including *Enhancer of split (E(Spl)), senseless (sens)*, and *patched (ptc)*, as well as several housekeeping genes that are not Mef2 targets (*Sandmann et al., 2006*), are not changed in *fh* mutants (*Figure 6—figure supplement 1A*). These data indicate that Mef2 is abnormally activated in *fh* mutants.

Next, we tested if the activated Pdk1/Mef2 pathway contributes to neurodegeneration of *fh* mutants. We reasoned that if Pdk1 and Mef2 are activated and toxic, their down-regulation may suppress degeneration. Consistent with our hypothesis, RNAi mediated knockdown of *Pdk1* suppresses the ERG and morphological defects in *fh* mutants (*Figure 6B* and *Figure 6—figure supplement 1B*). Furthermore, knock down of *Mef2* by two independent RNAi lines suppresses neurodegeneration of *fh* mutants (*Figure 6B* and *Figure 6—figure supplement 1B*), suggesting that activated Pdk1 and Mef2 are toxic and contribute to the degeneration. To determine if up-regulation of *Mef2* in wild type photoreceptors is sufficient to induce neurodegeneration, we ectopically expressed *Mef2* in the adult eye using Rh1-Gal4. Overexpression of *Mef2* is sufficient to cause photoreceptor degeneration (*Figure 6C*). Furthermore, flies with neuronal expression of *Mef2 (nSyb>Mef2)* die within a week, instead of 70–80 days of the wild type flies (data not shown). These data suggest that abnormal activation of Mef2 in neuron is sufficient to trigger its demise.

To assess if a similar pathway is affected in vertebrate cells, we tested if sphingolipids can induce Mef2 activation in Neuro-2A cells. dhSph, the sphingolipid that is increased more than 10 fold in fly *fh* mutants (*Figure 5A*), induces luciferase signals of a Mef2 reporter in a concentration dependent manner (*Figure 6D*). In addition, the increased luciferase signal is suppressed by adding the PDK1 inhibitor GSK2334470 (*Figure 6D*). This data indicate that dhsph activates Mef2 through PDK1, consistent with what we observed in the fly. Collectively, these data suggest that sphingolipids act as signaling molecules to activate the Pdk1/Mef2 pathway and that this pathway contributes to photoreceptor degeneration in *fh* mutants.

## Discussion

Here, we report the isolation of the first severe loss of function allele of *fh* in *Drosophila*. This allowed us to identify a novel molecular pathway that contributes to neurodegeneration. Loss of *fh* in *Drosophila* causes an age dependent degeneration of photoreceptors in mutant clones. Mutant photoreceptors display abnormal mitochondrial cristae morphology, reduced ETC CI activity, and impaired ATP production. However, we do not observe an increase in ROS. Our data indicate that Rh1 trafficking defects caused by mitochondria dysfunction lead to an activity dependent degeneration in *fh* mutants. However, the accumulation of $Fe^{2+}$ and/or $Fe^{3+}$ stimulates sphingolipid synthesis

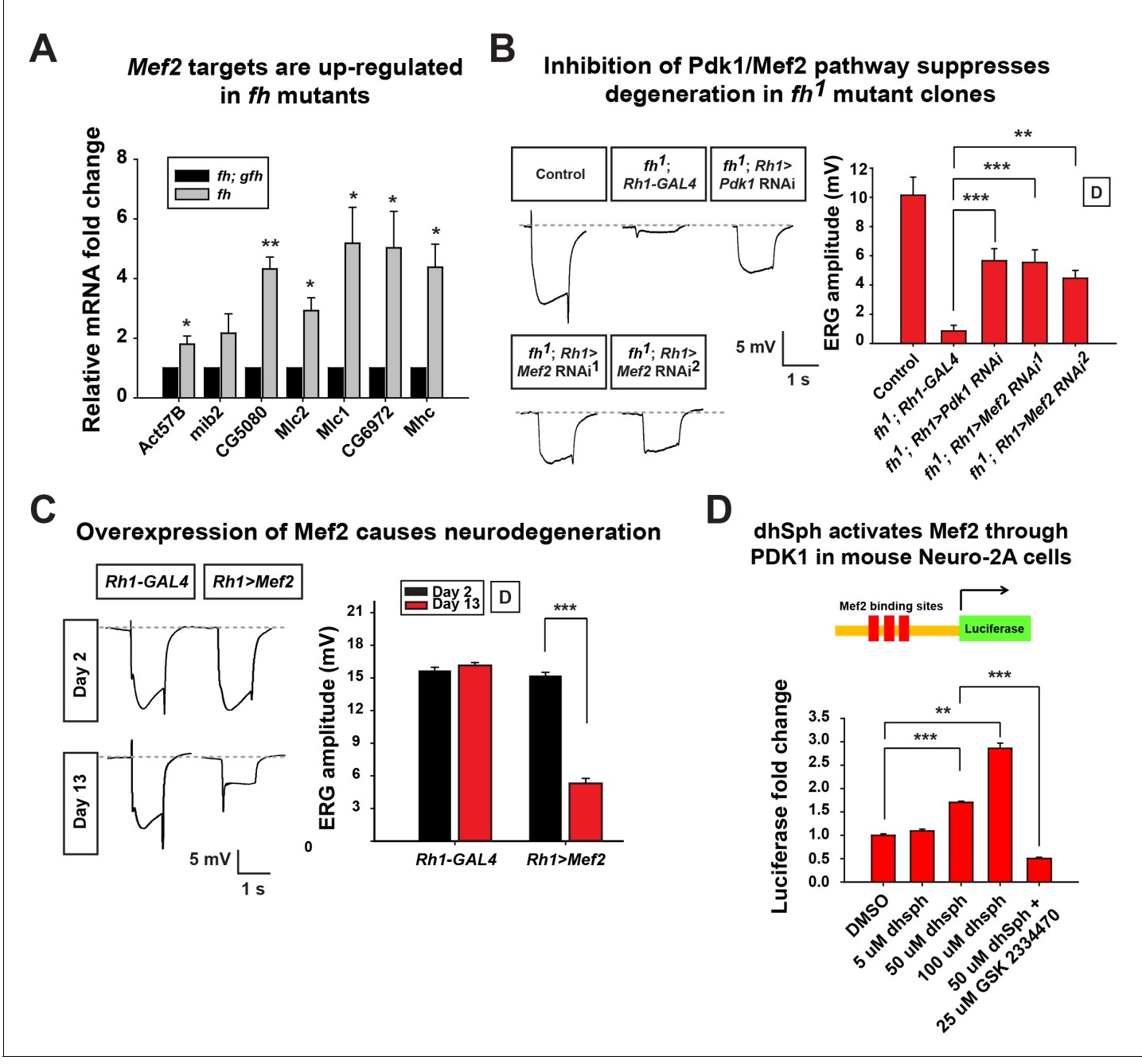

**Figure 6.** Activated Pdk1/Mef2 pathway causes degeneration in *fh* mutants. (**A**) mRNA levels of Mef2 downstream targets in *fh* mutants and rescued animals (*fh; gfh*). n = 3. (**B**) Day 3 ERG of control (*y w FRT19A; Rh1-GAL4*) and *fh* mutants in which an RNAi against Pdk1 or Mef2 is expressed. (**C**) ERG of flies with Mef2 overexpression in the eye. (**D**) Mef2 luciferase reporter assay in Neuro-2A cells with different concentration of dhSph treatment. GSK 2334470, a PDK1 inhibitor. n = 4. Data are presented as mean ± SEM. *p<0.05. **p<0.01. ***p<0.001, Student's t-test. D, flies are raised in dark.

The following figure supplement is available for figure 6:

**Figure supplement 1.** Control of qRT-PCR.

which in turn activates the Pdk1/Mef2 pathway to cause an activity independent degeneration. Hence, mitochondrial dysfunction, and iron toxicity through activation of the sphingolipid/Pdk1/Mef2 pathway, synergistically contribute to the demise of neurons in *fh* mutants. The data indicate

that ectopic activation of the transcription factor *Mef2* induced by loss of *FXN* may play a role in FRDA.

Elevated ROS levels, either through impaired ETC CI or CIII, or iron associated Fenton reaction, have been proposed to be the major driving force of disease pathogenesis in FRDA (*Bayot et al., 2011*; *Santos et al., 2010*). However, we do not observe increased ROS levels in *fh* mutants (*Figure 2D*). In addition, overexpression of ROS scavengers or feeding antioxidant AD4 does not suppress degeneration (*Figure 2E* and *Figure 2—figure supplement 1E*). The lack of increased ROS is surprising as mitochondrial mutants in *Drosophila* that exhibit a severely reduced CI activity typically show elevated ROS levels (*Owusu-Ansah et al., 2008*; *Zhang et al., 2013*). Numerous studies using different animal models or cell lines have established that a partial or complete loss of *FXN* leads to different degrees of defects in the ETC. Importantly, several reports have also documented that loss of *FXN* leads to an hypersensitivity to ROS, rather than elevated ROS levels (*Babcock et al., 1997*; *Llorens et al., 2007*; *Macevilly and Muller, 1997*; *Seznec et al., 2005*; *Shidara and Hollenbeck, 2010*). Consistent with our data, overexpression of ROS scavengers in an RNAi mediated *fh* knock down did not improve viability or cardiac function, and no ROS could be detected (*Anderson et al., 2005*; *Shidara and Hollenbeck, 2010*; *Tricoire et al., 2014*). Similarly, loss of *Fxn* in mice does not lead to an increase in ROS (*Puccio et al., 2001*; *Seznec et al., 2005*), although elevated ROS levels in another mouse model have been documented (*Al-Mahdawi et al., 2006*). Combined, these data suggest that ROS is not a prominent player in model organisms and a clinical trial designed to suppress ROS have not provided compelling data (*Santhera Pharmaceuticals, 2010*).

Iron accumulation in mitochondria was first described in *yeast frataxin homolog* mutants in yeast (*Babcock et al., 1997*). Although iron deposits have been reported in cardiac muscles of *Fxn* deficiency mice and FRDA patients, iron accumulation in the nervous system has been reported by some (*Boddaert et al., 2007*; *Koeppen et al., 2009*) but not by others (*Puccio et al., 2001*; *Simon et al., 2004*; *Solbach et al., 2014*). More importantly, the role of iron in the pathophysiology of FRDA has not yet been determined, and mitochondrial defects are thought to presage iron accumulation in mammals (*Puccio et al., 2001*). It has been argued that $Fe^{3+}$ accumulations are a non-reactive, intra-mitochondrial precipitates (*Seguin et al., 2010*; *Whitnall et al., 2012*). Finally, others proposed that iron accumulation is toxic as it induces the Fenton's reaction and promotes ROS production. We find that in *Drosophila fh* mutants, both $Fe^{2+}$ and $Fe^{3+}$ accumulate in the nervous system (*Figure 3C and D*). As we observed no increased ROS, we argue that $Fe^{2+}$ and $Fe^{3+}$ accumulation is toxic because: 1) the lethality is enhanced when iron is increased; 2) reduction of iron in food suppresses the progression of photoreceptor degeneration; and 3) overexpression of Ferritin delays the demise of neurons. Taken together, we argue that iron accumulation in neurons is toxic and plays a primary role in the pathogenesis in *fh* mutants.

Our data indicate that iron accumulation enhances the synthesis of sphingolipid, and that the excess amount of sphingolipids is toxic and promotes neurodegeneration in *fh* mutants. How iron accumulation leads to an increase in sphingolipid synthesis is unknown. Although we show that $Fe^{2+}$ accumulates in mitochondria of the larval muscle and at the NMJ in *fh* mutants, it is still not clear whether iron accumulates in other cellular organelles. The *de novo* synthesis of sphingolipids occurs in the ER, and mitochondria have close contacts with ER at the MAMs (mitochondria-associated endoplasmic reticulum membrane). Mitochondrial or ER localized iron may promote the enzymatic function of Lace or other proteins in the ER to enhance sphingolipid synthesis. However, treatment with Myriocin which targets Lace to decrease sphingolipid synthesis or down-regulation of *lace* via RNAi mediated knockdown suppresses degeneration in *fh* mutants (*Figure 5B and C*), suggesting that increased sphingolipid metabolites contribute to toxicity. These data are also in agreement with the observation that overexpression of *Sk2 (sphingosine kinase 2)* or feeding dhSph-1P to wild type flies promotes the degeneration of photoreceptors (*Yonamine et al., 2011*).

An increase in sphingolipids may affect various processes, including membrane integrity, lipid metabolism, and signal transduction (*Brown and London, 2000*; *Hannun and Obeid, 2008*; *Worgall, 2008*). We found that down-regulation of *Pdk1* or *Mef2* in mutant photoreceptor delays neurodegeneration in *fh* mutants, while overexpression of *Mef2* in photoreceptor promotes the demise of neurons (*Figure 6B and C*). It has been reported that Sph or dhSph, two sphingolipids that are elevated in *fh* mutants, directly induce autophosphorylation of PDK1 or SGK in an in vitro kinase assay (*Caohuy et al., 2014*; *King et al., 2000*). Although there is no evidence that PDK1 or SGK can directly activate Mef2, our data clearly show that Mef2 activity is up-regulated by elevating dhSph in

the medium of cultured Neuro-2A cells, and this increased activity is suppressed by a PDK1 inhibitor (*Figure 6D*), indicating that Mef2 can be activated by PDK1 or a downstream substrate of PDK1 (*Mora et al., 2004*).

The MEF2 family includes four vertebrate genes (*MEF2A–D*) that play a key role in muscle differentiation (*Molkentin et al., 1995*). However, the MEFs are also expressed in neurons where they are involved in neuronal development (*Akhtar et al., 2012*; *Leifer et al., 1994*; *Lyons et al., 1995*; *Mao et al., 1999*). The MEFs are known to regulate the expression of numerous target genes, including genes required for muscle differentiation, immediate-early genes, neuronal-activity-regulated genes, as well as genes involved in energy storage and immune response (*Clark et al., 2013*; *Dietrich, 2013*). In flies, ectopic expression of Mef2 in ectoderm is sufficient to induce nearly half of its predicted downstream targets that are typically expressed in muscles (*Sandmann et al., 2006*). Hence, its unusual activity may trigger neurodegeneration via the activation of some of its downstream targets. Importantly, it has been shown that the up-regulation of Mef2 is implicated in the cardiac hypertrophy and dilation (*Molkentin and Markham, 1993*; *Xu et al., 2006*), which is the major cause of death in FRDA patients. In sum, our data indicate that reducing the levels of sphingolipid synthesis with Myriocin, or antisense strategy against enzymes involved in sphingolipid synthesis, or reducing the levels of *Pdk1* or *Mef2*, are options that should be explored in FRDA models.

## Materials and methods

### Fly strains and genetics

Flies were obtained from the Bloomington Drosophila Stock Center at Indiana University (BDSC) unless otherwise noted. The stocks were routinely maintained at room temperature. For genetic interaction experiments, flies were raised at 28°C to enhance the GAL4 activity. For all the larval experiments, flies were allowed to lay eggs for 48 hr on grape juice plates with yeast paste. Hemizygous mutant larvae and wild type controls were isolated by GFP selection at the first instar phase and transferred to standard fly food for the duration of their development. To create mosaic mutant clones in the adult eye, *y w FRT19A* and *y w fh FRT19A/FM7c, Kr>GFP* females were crossed with *y w GMR-hid FRT19A; ey-GAL4 UAS-FLP* or *y w GMR-hid l(1)cl FRT19A/FM7c, Kr>GFP; eyFLP Rh1-GAL4/CyO* flies.

The following strains were used to generate fly stocks in this study:

*y w FRT19A* (*Haelterman et al., 2014*; *Yamamoto et al., 2014*)
*y w fh$^1$ FRT19A/FM7c, Kr>GFP* (*Haelterman et al., 2014*; *Yamamoto et al., 2014*)
*y w fh$^1$ FRT19A/FM7c, Kr>GFP; genomic-fh* (*Haelterman et al., 2014*; *Yamamoto et al., 2014*)
Df(1)BSC537, w$^{1118}$/FM7h/Dp(2;Y)G, P{hs-hid}Y
y w GMR-hid FRT19A; ey-GAL4 UAS-FLP
*y w GMR-hid l(1)cl FRT19A/FM7c, Kr>GFP; eyFLP Rh1-GAL4/CyO* (*Jaiswal et al., 2015*)
w$^1$; P{UAS-fh.A}1 (*Anderson et al., 2005*)
w$^1$; P{UAS-Cat.A}2 (*Anderson et al., 2005*)
w$^1$; P{UAS-Sod.A}B37 (*Anderson et al., 2005*)
w$^1$; P{UAS-Sod2.M}UM83 (*Anderson et al., 2005*)
P{rh1-GAL4}3, ry$^{506}$
P{Gal4-da.G32} UH1
y$^1$ w$^*$; P{nSyb-GAL4.S}3
w; repo-GAL4/TM3,Sb
w$^*$; P{w$^{+mC}$ bmm$^{Scer\UAS}$ = UAS-bmm} (*Gronke et al., 2005*) (gift from Ronald Kühnlein)
w*; UAS-Lip4/CyO (*Liu et al., 2015*)
P{UAS-Mef2-HA} (*Sandmann et al., 2006*) (gift from Eileen Furlong)
P{UAS-Fer1HCH} P{UAS-Fer2LCH} (*Missirlis et al., 2006, 2007*) (gift from Hermann Steller)
P{UAS-Fer1HCH.mut} P{UAS-Fer2LCH} (*Missirlis et al., 2006, 2007*) (gift from Hermann Steller)
P{UAS-Fer3HCH} (*Missirlis et al., 2006*) (gift from Patrik Verstreken)
w$^{1118}$; P{GawB}D42, P{UAS-mito-HA-GFP.AP}3 e$^1$/TM6B, Tb$^1$ (*Horiuchi et al., 2005*) (gift from Bill Saxton)
w*; cno$^3$ P{A92}puc$^{E69}$/TM6B, abdA-LacZ (*Ring and Martinez Arias, 1993*) (Drosophila Genetic Resource Center)

*ND42* RNAi: $y^1$ $sc^*$ $v^1$; P{TRiP.HMS00798}attP2 (*Owusu-Ansah et al., 2008*)
*lace* RNAi: P{KK102282}VIE-260B (*Ghosh et al., 2013*) (VDRC)
*Pdk1* RNAi: P{KK108363}VIE-260B (*Tokusumi et al., 2012*) (VDRC)
*Mef2* RNAi: $w^{1118}$; P{GD5039}v15549/TM3 and $w^{1118}$; P{GD5039}v15550 (*Clark et al., 2013*) (VDRC)

## ERG recording

ERG recordings were performed as described (*Verstreken et al., 2003*). Briefly, flies were glued on a glass slide. A recording electrode filled with 3M NaCl was placed on the eye, and another reference electrode was inserted into the fly head. Before the recording, photoreceptors were allowed to recover by keeping flies in the dark for 3 min. During the recording, a 1 s pulse of light stimulation was given, and the ERG traces of five to ten flies were recorded and analyzed by AXON -pCLAMP 8 software.

## Iron staining

To detect $Fe^{2+}$ levels, L3 larvae were dissected in ice-cold 0.25 mM $Ca^{2+}$ HL3 (*Stewart et al., 1994*) (70 mM NaCl, 5 mM KCl, 20 mM $MgCl_2$, 10 mM $NaHCO_3$, 5 mM trehalose, 5 mM HEPES, 115 mM sucrose, pH 7.2) and rinsed in HL3 with 1 mM $Ca^{2+}$. The animals were then incubated with 10 µM RPA or RPAC (Squarix Biotechnology, Germany) in 1 mM $Ca^{2+}$ HL3 for 20 min at room temperature in the dark. The animals were washed subsequently three times with ice-cold 1 mM $Ca^{2+}$ HL3 and then incubated in 1 mM $Ca^{2+}$ HL3 with 7 mM L-glutamic acid for 15 min at room temperature in the dark. The RPA or RPAC fluorescence was obtained with a Zeiss LSM 510 confocal microscope (Carl Zeiss) and quantified by Image J.

To evaluate $Fe^{3+}$ levels, L3 larval brains were dissected and fixed in 3.7% formaldehyde for 20 min. Samples were quickly washed three times with 0.4% Triton-PBS and incubated with Perls solution (1% $K_4Fe(CN)_6$ and 1% HCl in PBT) for 5 min. After 3 quick washes in PBT, samples were incubated with DAB solution (10 mg DAB with 0.07% $H_2O_2$ in 0.4% Triton-PBS) to enhance the signal. The brains were then quickly washed three times with 0.4% Triton-PBS and mounted. The brain images were obtained with a Leica MZ16 stereomicroscope equipped with Optronics MicroFire Camera.

## ROS levels analysis

L3 larvae were dissected in ice-cold 0.25 mM $Ca^{2+}$ HL3 and rinsed with 1 mM $Ca^{2+}$ HL3. The animals were then incubated with 10 µM dihydroethidium (DHE, Sigma-Aldrich) in 1 mM $Ca^{2+}$ HL3 for 20 min at room temperature in the dark. The samples were quickly washed three times with 1 mM $Ca^{2+}$ HL3 with 7 mM L-glutamic acid, and the DHE fluorescence in larval brains were live imaged by Zeiss 510 confocal microscope (Carl Zeiss) and quantified by Image J.

## Low iron food preparation and drug administration

Low iron food was made using 10% dry yeast, 10% sucrose, and 0.6% agar in w/v in water. The solution was microwaved and dried at room temperature, divided into vials, and stored at 4°C for further use. To increase iron levels in the food, ammonium iron (III) citrate (Sigma-Aldrich) was added to a final concentration of 1 mM or 10 mM in the low iron food. To test if inhibition of sphingolipid synthesis suppresses neurodegeneration in *fh* mutants, myriocin from *Mycelia sterilia* (Sigma-Aldrich) was dissolved in regular fly food (molasses based) or low iron food, and vials were then stored immediately at −20°C for future use. The flies were transfer to fresh food vials every two days.

## Molecular cloning

For the *fh* genomic rescue construct, a DNA fragment containing *Drosophila* X: 9,147,070..9,150,371 was retrieved from a 20 kb P[acman] construct that covers the entire *fh* genomic locus (clone CH322-14E18 from BACPAC Resources Center) (*Venken et al., 2009*). The *fh* genomic sequence was then subcloned into p{w⁺}attB using KpnI and NotI sites. The construct was then injected into *y w ΦC31; VK33* embryos, and transgenic flies were selected.

To generate pUAST-attB-UAS-hFXN construct, the primers 5'- TCCGAATTCGCCACCATG TGGACTCTCGGGCGCCGCGC-3' and 5'-GCGGTACCTCAAGCATCTTTTCCGGAATAGGC-3' were

used to clone the full length *FXN* cDNA from pcDNA-hFXN-HA, a gift from Gino Cortopassi (*Shan et al., 2007*). The PCR product was then subcloned into the pUAST-attB vector. The construct was then injected into *y w ΦC31;VK33* or *y w ΦC31;VK37* embryos, and the transgenic flies were selected.

## Immunofluorescence staining

For whole mount eye staining, we dissected fly heads in cold PBS and fixed with 4% paraformaldehyde at 4°C overnight. The retinas were then dissected and fixed for an additional 30 min. For larval ventral nerve cord staining, L3 instar larvae were dissected and fixed with 3.7% formaldehyde for 20 min. The antibodies were used at the following concentrations: mouse anti-Na/K ATPase (α5, Developmental Studies Hybridoma Bank (DSHB)), 1:200; rat anti-Elav (7E8A10, DSHB), 1:500; mouse anti-Rh1 (4C5, DSHB), 1:100; mouse anti-V5 (R96025, Invitrogen), 1:1000; rabbit anti-HRP (323-005-021, Jackson ImmunoResearch), 1:1000; chicken anti-GFP (ab13970, abcam), 1:1000; rabbit anti-β-Galactosidase (Cappel), 1:1000; mouse anti-DLG (4F3, DSHB), 1:200; Alexa 488-conjugated phalloidin (Invitrogen), 1:100; Alexa 405-, Alexa 488-, Cy3-, or Cy5-conjugated secondary antibodies (Jackson ImmunoResearch), 1:250. Samples were then mounted in Vectashield (Vector Laboratories) before being analyzed under a confocal microscope. All the confocal scans were acquired with a confocal microscope (model LSM 510 or LSM 710; Carl Zeiss) and processed using LSM Image Browser (Carl Zeiss) and Photoshop (Adobe).

The Nile red staining was performed as described (*Liu et al., 2015*). Briefly, the retina was dissected and fixed in 3.7% formaldehyde. The samples were then incubated for 10 min at 1:1,000 dilution of PBS with 1 mg/ml Nile Red (Sigma). The retina was then washed with PBS and mounted in Vectashield (Vector Laboratories). Images were obtained with a Zeiss LSM 510 confocal microscope.

## Mitochondrial functional assay

Mitochondria were extracted as previously described (*Zhang et al., 2013*). Mitochondrial ETC enzymatic activity assay and aconitase activity assay were performed as previously described (*Zhang et al., 2013*). To determine ADP/ATP ratio, L3 larvae were collected and the ratio was determined with an ADP/ATP Ratio Assay Kit (ab65313, abcam). Briefly, 5–10 L3 instar larvae were collected and frozen with liquid nitrogen. The samples were then homogenized and the ADP/ATP ratio was measured following manufacturer's protocol. The luminescence was measured by Synergy 2 Microplate Reader (BioTek).

## Transmission electron microscopy (TEM)

*Drosophila* retina ultrastructure was imaged following standard electron microscopy procedures using a Ted Pella Bio Wave processing microwave with vacuum attachments. Briefly, fly heads were dissected and fixed at 4°C in fixative (2% paraformaldehyde, 2.5% glutaraldehyde, 0.1 M sodium cacodylate, and 0.005% $CaCl_2$, pH 7.2) overnight. The samples were then postfixed in 1% $OsO_4$, dehydrated in ethanol and propylene oxide, and then embedded in Embed-812 resin (Electron Microscopy Sciences) under vacuum. Photoreceptors were then sectioned and stained in 1% uranyl acetate and saturated lead nitrate. TEM images of photoreceptor sections were taken using a JEOL JEM 1010 transmission electron microscope at 80 kV with an AMT XR-16 mid-mount 16 mega-pixel digital camera.

## Mass spectrometry

ESI/MS/MS analysis of endogenous sphingosine bases and ceramide species (C14- and C16-Sphingoid Base) were performed on a Thermo_Fisher TSQ Quantum triple quadrupole mass spectrometer, operating in a Multiple Reaction Monitoring (MRM) positive ionization mode, using modified version (*Bielawski et al., 2009*). Briefly, whole larvae were fortified with the internal standards (ISs: C17 base D-erythro-sphingosine (17CSph), C17 sphingosine-1-phosphate (17CSph-1P), N-palmitoyl-D-erythro-C13 sphingosine (13C16-Cer) and heptadecanoyl-D-erythro-sphingosine (C17-Cer)), and extracted with ethyl acetate/iso-propanol/water (60/30/10 v/v) solvent system. After evaporation and reconstitution in 100 µl of methanol samples were injected on the HP1100/TSQ Quantum LC/MS system and gradient eluted from the BDS Hypersil C8, 150 × 3.2 mm, 3 µm particle size column, with 1.0 mM methanolic ammonium formate / 2 mM aqueous ammonium formate mobile phase

system. Peaks corresponding to the target analytes and internal standards were collected and processed using the Xcalibur software system.

Quantitative analysis was based on the calibration curves generated by spiking an artificial matrix with the known amounts of the target analyte synthetic standards and an equal amount of the internal standards (ISs). The target analyte/IS peak areas ratios were plotted against anlyte concentration. The target analyte/IS peak area ratios from the samples were similarly normalized to their respective ISs and compared to the calibration curves, using a linear regression model. Applying consistent mass spectral conditions of Collision Assistant Dessociation (CAD); 35 eV and Electron Spray Ionization (ESI) all sphingoid bases and related ceramides undergo uniform transition from initial molecular ion (M+1) to the respective sphingoid backbone secondary ions. Consequently, calibration curves, generated from authentic standards of the typical, 18C-sphingosine and ceramides, can be used for quantitation of other, e.g. 20C- counterparts. The levels of sphingolipid species are normalized to the phosphate amount of the samples. Although absolute concentrations determined for compounds without authentic standards (20C-LCB derivatives) may not be precise, due to possible differences in instrument response for 18C- and 20C-LCB related compounds, for comparative study, where changes in sphingolipids level rather than absolute concentration, are most important this indirect methodology provide reliable results.

## Reverse transcription–quantitative polymerase chain reaction (PCR)

Total RNA from the fly larval brains were extracted by Trizol RNA Isolation Reagents (Thermo Fisher Scientific) follow the manufacturer's instructions. The cDNA was synthesized by High-Capacity cDNA Reverse Transcription Kit (Applied Biosystems). Real-time PCR was carried out using iQ SYBR Green Supermix (Bio-Rad) and performed in a thermal cycler (iCycler; Bio-Rad Laboratories). The data were collected and analyzed using the optical module (iQ5; Bio-Rad Laboratories). The following primer pairs are used (5'to 3'): RP49 forward (control primer), TCCTACCAGCTTCAAGATGAC; RP49 reverse (control primer), CACGTTGTGCACCAGGAACT; Act57B forward, TTGAGACCTTCAACTCGCCC; Act58B reverse, CCATCACCGGAGTCCAGAAC; mib2 forward, CGCCAGAAAACACTGTCGTG; mib2 reverse, GACGAACTCCAACTTGAGCATTA; CG5080 forward, CGCCCTCTCCAATTAGTTC TCC; CG5080 reverse, CAGCGACTGGATAGTTCCGC; Mlc2 forward, TCGGGTCCGATCAACTTCAC ; Mlc2 reverse, ATTTCGCGGAATTTGTCACCG; Mlc1 forward, CCGAGGATGATGAAGGATTT; Mlc1 reverse, CTGGTCTGTCACACATTCTGG; CG6972 forward, AGGGATCACACACTGATGAACT; CG6972 reverse, CAACAGCCATTCGGAGGGAC; Mhc forward, ATCAATCCTTACAAGCGTTACCC; Mhc reverse, CCGTCAGAGATGGCGAAAATATG; E(Spl) forward, ATGGAATACACCACCAA-GACCC; E(Spl) reverse, GGCGACAAGTGTTTTCAGGTT; sens forward, TACTGTGGCCCCAA TTTTTGT; sens reverse, AAGGCAAAGTCACGATCCCG; ptc forward, GGATCTTTACATACGCAC-CAGC; ptc reverse, GGACTGGAATACTGATCGCAG;

## Mef2 Luciferase reporter assay

Neuro-2A cells were seed on 12 well plates, and each well was transfected with 3XMef2-luc reporter (Addgene) and Renilla control pRL-TK vector (gift from Huda Zoghbi). After two days, DMSO, DL-Dihydrosphingosine (Sigma), or GSK2334470 (gift from Huda Zoghbi) was added into the medium. Cells were cultured one more day with drug treatment. The luciferase assay was then performed by Dual-Luciferase Reporter Assay System (Promega) and followed manufacturer instruction.

## Acknowledgements

We thank Karen Schulze for comments; Fanis Missirlis and Hermann Steller (ferritin transgenic flies); Patrik Verstreken (UAS-Fer3HCH); Eileen Furlong (UAS-mef2); Gino Cortopassi (pcDNA-hFXN-HA); Huda Zoghbi (pRL-TK vector and GSK2334470); Addgene, the Bloomington Drosophila Stock Center, the Vienna Drosophila Resource Center, and the Kyoto Drosophila Genetic Resource Center for providing stocks and reagents. We thank the Lipidomics Core at the Medical University of South Carolina for sphingolipid analysis. We thank the BCM Intellectual and Developmental Disabilities Research Center (IDDRC) confocal microscopy core, which were supported by the Eunice Kennedy Shriver National Institute of Child Health & Human Development (1U54 HD083092). SY is supported by the Jan and Dan Duncan Neurological Research Institute at Texas Children's Hospital. We acknowledge support from the Friedreich's Ataxia Research Alliance, NIH (1RC4GM096355), the

Robert A and Renee E. Belfer Family Foundation, the Huffington Foundation, and Target ALS to HJB. HJB is an Investigator of the Howard Hughes Medical Institute.

# Additional information

## Competing interests

HJB: Reviewing editor, *eLife*. The other authors declare that no competing interests exist.

## Funding

| Funder | Grant reference number | Author |
|---|---|---|
| Friedreich's Ataxia Research Alliance | | Kuchuan Chen<br>Guang Lin<br>Hugo J Bellen |
| Howard Hughes Medical Institute | | Lita Duraine<br>Manish Jaiswal<br>Hugo J Bellen |
| Texas Children's Hospital | Jan and Dan Duncan Neurological Research Institute | Shinya Yamamoto |
| National Institutes of Health | 1RC4GM096355 | Hugo J Bellen |
| Robert A and Renee E. Belfer Family Foundation | | Hugo J Bellen |
| Huffington Foundation | | Hugo J Bellen |
| Target ALS | | Hugo J Bellen |
| Eunice Kennedy Shriver National Institute of Child Health and Human Development | 1U54 HD083092 | Hugo J Bellen |

The funders had no role in study design, data collection and interpretation, or the decision to submit the work for publication.

## Author contributions

KC, TS-YH, Conception and design, Acquisition of data, Analysis and interpretation of data, Drafting or revising the article, Contributed unpublished essential data or reagents; GL, Conception and design, Acquisition of data, Analysis and interpretation of data, Drafting or revising the article; NAH, Conception and design, Drafting or revising the article; TL, MJ, SY, Acquisition of data, Drafting or revising the article; ZL, LD, Acquisition of data, Analysis and interpretation of data, Drafting or revising the article; BHG, HJB, Conception and design, Analysis and interpretation of data, Drafting or revising the article; MNR, Conception and design, Drafting or revising the article, Contributed unpublished essential data or reagents

## Author ORCIDs

Brett H Graham, http://orcid.org/0000-0001-8451-8154
Hugo J Bellen, http://orcid.org/0000-0001-5992-5989

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
