## [Decision Letter]

Thank you for submitting your article "Loss of Frataxin causes iron toxicity, induces sphingolipid synthesis and PDK1/Mef2 pathway to trigger neurodegeneration" for consideration by *eLife*. Your article has been favorably evaluated by a Senior editor and three reviewers, one of whom, J. Paul Taylor, is a member of our Board of Reviewing Editors.

The reviewers have discussed the reviews with one another and the Reviewing Editor has drafted this decision to help you prepare a revised submission.

Summary:

Loss of the nuclear-encoded mitochondrial protein Frataxin (FXN) causes Friedreich's ataxia (FRDA), characterized by progressive neurodegeneration impacting the dorsal root ganglia, sensory peripheral nerves, corticospinal tracts, and dentate nuclei of the cerebellum. Frataxin is required for assembly of iron-sulfur clusters, which are essential components of metalloproteins of the electron transport chain (ETC). A longstanding hypothesis in the FRDA field is that loss of Frataxin leads to accumulation of free iron and impairment of the ETC resulting in excess reactive oxygen species that causes neurodegeneration. However, this hypothesis has been challenged by inconsistent findings in model systems and patients, as well as the lack of efficacy in antioxidant therapy in multiple clinical trials.

The current manuscript describes the characterization of a mutant allele of the orthologous *Drosophila* FXN gene (*fh*) that was identified in an unbiased forward genetic EMS screen for neurodegeneration mutants. These animals exhibit iron accumulation but no increased ROS. Rather, these animals show significant up-regulation of sphingolipid synthesis, resulting in activation of a signaling pathway that includes 3-phosphoinositide dependent protein kinase-1 (Pdk1) and myocyte enhancer factor-2 (Mef2). The role of these factors in driving neurodegeneration is established with genetic pharmacological approaches in fly. Evidence is presented that depletion of endogenous FXN in mouse cortical neurons results in iron accumulation and activation of Pdk1/Mef2, thus beginning to address the physiological relevance of this pathogenic mechanism in mammals.

Essential revisions:

1) This is a meticulously conducted study that features clear and persuasive data with respect to the findings in *Drosophila*. While some of the findings here have been reported previously (such as lack of increased ROS in Frataxin LOF flies), much of what is presented here is novel and, moreover, the study provides compelling evidence suggesting that iron-mediated toxicity initiates a signaling cascade that culminates in increased sphingolipid biosynthesis and activation of the Pdk1/Mef2 pathway, with deleterious consequences. Important unanswered questions include, what is the mechanism whereby iron accumulation results in increased sphingolipid synthesis? Is this a generic consequence of iron overload? And what is the mechanism whereby increased sphingolipid synthesis activates of PDK1 and Mef2? It may not be practical to thoroughly answer these questions in the current manuscript, but they should be acknowledged and explored in the Discussion.

2) The evidence that RNAi-mediated depletion of Pdk1 and Mef2 RNAi suppresses neurodegeneration in *fh* mutant is supported by electrophysiology data, but would be strengthened by evidence that the morphological defect in the retina is also suppressed. Is this suppression specific to *fh* mutants? Where possible, multiple RNAi lines should be used.

3) Whereas the reviewers feel that the fly data is rigorous, thorough and compelling, the mouse data was viewed as preliminary and comparatively superficial. Convincing evidence that the mechanism elucidated in fly is also operational in mouse would be important, but we are not persuaded that sufficient evidence has been provided. We do not feel that the mouse data is essential to the current manuscript and suggest that it be removed. If the phenotype of the FXN-depleted mice can be more thoroughly characterized it may be appropriate for follow up publication as "Research Advance." e*Life* Research Advances do not have to have the scope expected for stand-alone papers, but significantly add to the message of the previous paper.

The specific concerns relating to the mouse data were:

The reviewers did not feel that significant Fe^2+^ accumulation in the brain was convincingly documented. The evidence consists solely of low power micrographs of unclear cortical regions scored for loss of signal, without evidence that Frataxin expression is reduced in these same cells. Since the conclusion drawn here contradicts a prior report, the standard of evidence must be high. The assay employed was quenching of RPA signal, whereas the prior report was based on the well-established Perl's Prussian Blue stain. The authors' claim that RPA quenching is more sensitive to Fe^2+^ levels, but do not provide evidence of this or cite a reference. We failed to find evidence of this in the literature.

Do the mice show evidence of increased ROS?

Do the mice show increased synthesis of sphingolipids?

The evidence of activation of PDK1/Mef2 is limited and indirect, showing increased Western signal with an antibody for phospho-241-PDK1 and increased transcript levels for handpicked Mef2 target genes. Is overall expression of PDK1 increased, or only the phosphorylation level?

---

## [Author Response]

Essential revisions:

*1) This is a meticulously conducted study that features clear and persuasive data with respect to the findings in Drosophila. While some of the findings here have been reported previously (such as lack of increased ROS in Frataxin LOF flies), much of what is presented here is novel and, moreover, the study provides compelling evidence suggesting that iron-mediated toxicity initiates a signaling cascade that culminates in increased sphingolipid biosynthesis and activation of the Pdk1/Mef2 pathway, with deleterious consequences. Important unanswered questions include, what is the mechanism whereby iron accumulation results in increased sphingolipid synthesis? Is this a generic consequence of iron overload? And what is the mechanism whereby increased sphingolipid synthesis activates of PDK1 and Mef2? It may not be practical to thoroughly answer these questions in the current manuscript, but they should be acknowledged and explored in the Discussion.*

To address this issue in the Discussion, we added the following section:

“…How iron accumulation leads to an increase in sphingolipid synthesis is unknown. Although we show that Fe^2+^ accumulates in mitochondria of the larval muscle and at the NMJ in *fh* mutants, it is still not clear whether iron accumulates in other cellular organelles. The de novo synthesis of sphingolipids occurs in the ER, and mitochondria have close contacts with ER at the MAMs (mitochondria-associated endoplasmic reticulum membrane). Mitochondrial or ER localized iron may promote the enzymatic function of Lace or other proteins in the ER to enhance sphingolipid synthesis.”

“…It has been reported that Sph or dhSph, two sphingolipids that are elevated in *fh* mutants, directly induce autophosphorylation of PDK1 or SGK in an in vitro kinase assay (King et al., 2000; Caohuy et al., 2014). Although there is no evidence that PDK1 or SGK can directly activate Mef2, our data clearly show that Mef2 activity is up-regulated by elevating dhSph in the medium of cultured Neuro-2a cells, and this increased activity is suppressed by a PDK1 inhibitor (Figure 6), indicating that Mef2 can be activated by PDK1 or a downstream substrate of PDK1 (Mora et al., 2004).”

2) The evidence that RNAi-mediated depletion of Pdk1 and Mef2 RNAi suppresses neurodegeneration in fh mutant is supported by electrophysiology data, but would be strengthened by evidence that the morphological defect in the retina is also suppressed. Is this suppression specific to fh mutants? Where possible, multiple RNAi lines should be used.

Knock down of *Pdk1* or *Mef2* by RNAi suppresses morphological defects in *fh* mutant clones and the images are now presented in Figure 6—figure supplement 1.

Whether this suppression is specific to *fh* mutants is still unknown. We are analyzing a phospholipase mutant (iPLA2, also named PARK14) that in contrast to *fh* mutants exhibits a decrease in sphingolipids. Knocking down Pdk1 or Mef2 in these mutants does not suppress the degenerative phenotype, suggesting that just reducing the levels of Pdk1 or Mef2 does not non-selectively suppress neurodegeneration. As we do not have other models with increased levels of sphingolipids, we cannot better test specificity.

For the Pdk1 RNAi line, we were not able to find another RNAi line. However, we used two different independent Mef2 RNAi lines to suppress degeneration of *fh* mutants. In addition, we also show that a PDK1 inhibitor (GSK2334470) suppresses the activation of Mef2 (Figure 6).

*3) Whereas the reviewers feel that the fly data is rigorous, thorough and compelling, the mouse data was viewed as preliminary and comparatively superficial. Convincing evidence that the mechanism elucidated in fly is also operational in mouse would be important, but we are not persuaded that sufficient evidence has been provided. We do not feel that the mouse data is essential to the current manuscript and suggest that it be removed. If the phenotype of the FXN-depleted mice can be more thoroughly characterized it may be appropriate for follow up publication as "Research Advance." eLife Research Advances do not have to have the scope expected for stand-alone papers, but significantly add to the message of the previous paper.*

*The specific concerns relating to the mouse data were:*

*The reviewers did not feel that significant Fe^2+^ accumulation in the brain was convincingly documented. The evidence consists solely of low power micrographs of unclear cortical regions scored for loss of signal, without evidence that Frataxin expression is reduced in these same cells. Since the conclusion drawn here contradicts a prior report, the standard of evidence must be high. The assay employed was quenching of RPA signal, whereas the prior report was based on the well-established Perl's Prussian Blue stain. The authors' claim that RPA quenching is more sensitive to Fe^2+^ levels, but do not provide evidence of this or cite a reference. We failed to find evidence of this in the literature.*

*Do the mice show evidence of increased ROS?*

*Do the mice show increased synthesis of sphingolipids?*

*The evidence of activation of PDK1/Mef2 is limited and indirect, showing increased Western signal with an antibody for phospho-241-PDK1 and increased transcript levels for handpicked Mef2 target genes. Is overall expression of PDK1 increased, or only the phosphorylation level?*

We removed the mouse data and will address these questions in the “Research Advance”.